# Diminishment of Nrf2 Antioxidative Defense Aggravates Nephrotoxicity of Melamine and Oxalate Coexposure

**DOI:** 10.3390/antiox10091464

**Published:** 2021-09-14

**Authors:** Chia-Fang Wu, Chia-Chu Liu, Yi-Chun Tsai, Chu-Chih Chen, Ming-Tsang Wu, Tusty-Jiuan Hsieh

**Affiliations:** 1Research Center for Environmental Medicine, Kaohsiung Medical University, Kaohsiung 807378, Taiwan; chiafangwu27@gmail.com (C.-F.W.); m8201055@hotmail.com (C.-C.L.); lidam65@yahoo.com.tw (Y.-C.T.); ccchen@nhri.edu.tw (C.-C.C.); 960021@cc.kmuh.org.tw (M.-T.W.); 2International Master Program of Translational Medicine, National United University, Miaoli 360301, Taiwan; 3School of Medicine, College of Medicine, Kaohsiung Medical University, Kaohsiung 807378, Taiwan; 4Department of Urology, Kaohsiung Medical University Hospital, Kaohsiung Medical University, Kaohsiung 807378, Taiwan; 5Department of Urology, Pingtung Hospital, Ministry of Health and Welfare, Pingtung City 900027, Taiwan; 6Divisions of Nephrology, Kaohsiung Medical University Hospital, Kaohsiung Medical University, Kaohsiung 807378, Taiwan; 7Institute of Population Health Sciences, National Health Research Institutes, Miaoli 350401, Taiwan; 8Ph.D. Program in Environmental and Occupational Medicine, College of Medicine, Kaohsiung Medical University, Kaohsiung 807378, Taiwan; 9Graduate Institute of Clinical Medicine, College of Medicine, Kaohsiung Medical University, Kaohsiung 807378, Taiwan; 10Department of Family Medicine, Kaohsiung Medical University Hospital, Kaohsiung Medical University, Kaohsiung 807378, Taiwan; 11Graduate Institute of Medicine, College of Medicine, Kaohsiung Medical University, Kaohsiung 807378, Taiwan; 12Department of Marine Biotechnology and Resources, College of Marine Sciences, National Sun Yat-Sen University, Kaohsiung 804201, Taiwan

**Keywords:** chronic kidney disease, urolithiasis, hydroxy-L-proline, no-observed-adverse-effect level, 8-oxo-2′-deoxyguanosine, Nrf2, OGG1

## Abstract

Chronic kidney disease (CKD) usually causes devastating healthy impacts on patients. However, the causes affecting the decline of kidney function are not fully revealed, especially the involvement of environmental pollutants. We have revealed that exposure to melamine, a ubiquitous chemical in daily life, is linked to adverse kidney outcomes. Hyperoxaluria that results from exposure to excessive oxalate, a potentially nephrotoxic terminal metabolite, is reportedly associated with CKD. Thus, we explored whether interaction of these two potential nephrotoxicants could enhance kidney injury. We established a renal proximal tubular HK-2 cell model and a Sprague–Dawley rat model of coexposure to melamine with sodium oxalate or hydroxy-L-proline to investigate the interacting adverse effects on kidneys. Melamine and oxalate coexposure enhanced the levels of reactive oxygen species, lipid peroxidation and oxidative DNA damage in the HK-2 cells and kidney tissues. The degrees of tubular cell apoptosis, tubular atrophy, and interstitial fibrosis were elevated under the coexposed condition, which may result from the diminishment of Nrf2 antioxidative capacity. To conclude, melamine and oxalate coexposure aggravates renal tubular injury via impairment of antioxidants. Accumulative harmful effects of exposure to multiple environmental nephrotoxicants should be carefully investigated in the etiology of CKD progression.

## 1. Introduction

Chronic kidney disease (CKD) is an urgent issue worldwide and usually causes devastating healthy impacts on patients [1]. However, the causes affecting the decline of kidney function are not fully revealed. Urolithiasis and kidney stones are risk factors for patients to develop CKD and end-stage renal disease (ESRD) [1,2,3]. Calcium oxalate is the most common composition (approximately 80%) of kidney stones, while uric acid is present in only ~9% of kidney stones [2]. Oxalate can be ingested from many foods and its potentially toxic terminal metabolite is eliminated primarily by the kidney [3,4,5]. Hyperoxaluria has been reported to be associated with CKD and ESRD [1,2,3,4,5].

Melamine is a widely used chemical found in many daily use products and ubiquitously presents in our environment. The contamination of melamine is more pervasive than originally thought [6,7]. Environmental melamine exposure can come from daily contact with melamine-containing products such as melamine tableware. We have reported that substantial amounts of melamine can migrate out of these products, especially at high temperatures or high acidity [8,9]. In addition, the contamination by melamine and its derivatives (cyanuric acid, ammeline, and ammelide) remains prevalent in daily food products including infant formula, milk, yogurt, cheese, butter, and bread [10]. Several studies have reported that melamine is detectable in most urine samples from the general population, indicating it may universally exist in human bodies [11,12,13,14,15].

Kidney is the most susceptible organ to melamine since 90% of melamine is excreted in the original form in urine after intake [16]. Our previously studies have found that environmental melamine exposure is linked to the adverse kidney outcomes, such as urolithiasis and kidney function deterioration in adults [17,18,19,20]. Since consuming high-content oxalate foods has been suggested to be a risk of CKD [1,3,5], we hypothesized that coexposure to the two common kidney toxicants, melamine, and oxalate, might aggravate their nephrotoxicity.

Our previous human proximal tubular cell studies have suggested that oxidative stress could be a key player in kidney injury resulting from melamine exposure [21]. In the human studies, we also found there were interrelationships among environmental melamine, oxidative stress, and early kidney injury in workers from melamine tableware factories and adult patients with calcium urolithiasis [22]. However, these in vitro and human studies still cannot provide solid mechanistic evidence to answer how chronic exposure to environmental concentrations of melamine could result in early kidney damage, and whether oxidative stress is the critical underlying mechanism.

Inflammation is a leading cause of CKD [23]. Nuclear factor kappa B (NF-κB) regulates the expression of many genes that play key roles in inflammation, apoptosis, and fibrosis during the progression of CKD [23]. Renal tubulointerstitial fibrosis is a consequence of excessive accumulation of extracellular matrix proteins that occurs in nearly every type of CKD [24]. Transforming growth factor beta 1 (TGF-β1) stimulates overexpression of extracellular matrix proteins, such as collagen IV, and is the key regulator of renal fibrosis [24]. Both inflammation and fibrosis can be caused by oxidative stress [25]. The nuclear factor–erythroid-2–related factor 2 (Nrf2) is a master regulator of many antioxidants and can be activated under oxidative stress to initiate cell defense via upregulation of antioxidant enzymes, such as NAD(P)H dehydrogenase quinone 1 (NQO1), catalase, heme oxygenase 1 (HO1), superoxide dismutase 1 (SOD1), and superoxide dismutase 2 (SOD2) [26,27,28,29].

The aims of current study were to clarify whether environmental concentrations of melamine and oxalate coexposure could aggravate nephrotoxicity and what could be the roles of oxidative stress and Nrf2 in the kidney injury induced by the coexposure using the HK-2 cell and murine models. Our results demonstrated that the coexposure resulted in lethal and sublethal kidney damage, particularly renal tubular injury, by diminishing Nrf2 function and increasing oxidative stress.

## 2. Materials and Methods

### 2.1. Cell Culture

Human renal proximal tubular cells (HK-2 cells; BCRC no. 60097; Bioresource Collection and Research Center, Taiwan) were cultured in Dulbecco’s Modified Eagle Medium (DMEM; catalog no. (cat.) 12320; Thermo Fisher Scientific, Inc., Waltham, MA, USA) supplemented with 10% fetal bovine serum (FBS; cat. 10081148; Thermo Fisher Scientific, Inc.), 100 U/mL penicillin, and 100 μg/mL streptomycin (cat. 15140122; Thermo Fisher Scientific, Inc.) in a humidified atmosphere of 95% air and 5% CO_2_ at 37 °C. When the cells were reached 70% confluence, they were synchronized overnight in DMEM containing 0.5% FBS before being stimulated. Then, the cells were stimulated with melamine (cat. 108-78-1; Sigma-Aldrich, Inc., Saint Louis, MO, USA) and/or sodium oxalate (SO; cat. 131706; PanReac AppliChem GmbH, Darmstadt, Germany) under the cultural condition in which DMEM containing 0.5% FBS.

### 2.2. Exposure Concentrations of Melamine and Oxalate

To mimic the environmental exposure, the concentration of melamine used to stimulate HK-2 cells was according to our previous study that found the urinary melamine concentrations of melamine manufacturers range from 0.438 to 1.252 μg/mL [30]. Thus, in this study, we exposed HK-2 cells to 1 μg/mL melamine. We used 0.01 mM SO to stimulate the cells and the final concentration of oxalate in the culture medium was 0.88 μg/mL. This concentration is lower than the urinary oxalate concentration that has been reported to increase the risk of kidney stone formation and CKD [3,31].

### 2.3. Nrf2 RNA Interference

The functions of Nrf2 were knocked down by a small interfering RNA (siRNA) method. Nrf2 siRNA (h) (cat. sc-37030; Santa Cruz Biotechnology, Santa Cruz, CA, USA) was transfected into HK-2 cells with FuGENE HD transfection reagent (cat. E2311; Promega Corporation, Madison, WI, USA). HK-2 cells were cultured in 24-well or 6-well plates until they reached approximately 60–70% confluence. To knock down Nrf2, 30 nM Nrf2 siRNA with transfection reagent was added to cells grown in DMEM containing 10% FBS, and the cells were incubated for 24 h. Then, the cells were used for the following experiments.

To observe protein expression, cells with or without Nrf2 siRNA treatment were synchronized in DMEM containing 0.5% FBS overnight. Then, the cells were stimulated with 1 μg/mL melamine and/or 0.01 mM SO for 6 or 24 h.

### 2.4. Detection of Oxidative Stress

We applied 5-(and-6)-chloromethyl-2′,7′-dichlorodihydrofluorescein diacetate (CM-H_2_DCFDA) reagent (cat. C6827; Thermo Fisher Scientific, Inc.) to detect the reactive oxygen species (ROS) produced from HK-2 cells. Mitochondrial ROS was measured using MitoTracker Red CMXRos (cat. M7512; Thermo Fisher Scientific, Inc.), a mitochondrial dye that enters live cells and produces fluorescence after the dye is oxidized by superoxide in mitochondria. Hoechst 33342 (cat. H1399; Thermo Fisher Scientific, Inc.), a cell membrane-permeable dye, was used to stain cell nuclei.

HK-2 cells with or without Nrf2 siRNA treatment were cultured in 24-well plates and synchronized in DMEM containing 0.5% FBS overnight. Then, the cells were stimulated with 1 μg/mL melamine and/or 0.01 mM SO. After 6 h, CM-H_2_DCFDA (5 μM), MitoTracker Red CMXRos (0.25 μM), and Hoechst 33342 (2 μM) were added, and the cells were incubated for 30 min. Then, the cells were washed three times with 1× phosphate-buffered saline (PBS) and observed under an EVOS M5000 Imaging System fluorescence microscope (Thermo Fisher Scientific, Inc.).

The intensity and area of CM-H_2_DCFDA, MitoTracker Red CMXRos, and Hoechst 33342 fluorescence staining were analyzed using ImageJ software (a public domain Java image processing and analysis program developed at the National Institutes of Health, USA; https://imagej.nih.gov/ij/, accessed on 28 February 2019). To quantify ROS, the intensity of CM-H_2_DCFDA and MitoTracker Red CMXRos fluorescence was calibrated by dividing it with the area of Hoechst 33342 fluorescence.

### 2.5. Observation of Apoptosis

Hoechst 33342 was used to monitor nuclear morphology and thus to monitor apoptosis. HK-2 cells with or without Nrf2 siRNA treatment were cultured in 24-well plates and synchronized in DMEM containing 0.5% FBS overnight. Then, the cells were stimulated with 1 μg/mL melamine and/or 0.01 mM SO. After 6 or 24 h, Hoechst 33342 (2 μM) was added, and the cells were incubated for 30 min. Then, the cells were washed 3 times with PBS and observed under an EVOS M5000 Imaging System fluorescence microscope (Thermo Fisher Scientific, Inc., Waltham, MA, USA). Apoptotic cells were identified by chromatin condensation (presentation of enhanced blue fluorescence) and nuclear fragmentation.

### 2.6. Animal Experiment

The World Health Organization (WHO) originally recommended human tolerable daily intake (TDI) of melamine was based on the fact that no substantial renal toxicity was observed after 13 weeks of exposure to melamine at a dose of 63 mg/kg/day, the no-observed-adverse-effect level (NOAEL) for bladder calculi [32,33]. Thus, in the in vivo model, we used the NOEL of melamine as the dosage to test the effect of coexposure with oxalate. The chemical chosen to be coexposed with melamine was 4-hydroxy-L-proline (HLP; cat. 51-35-4; Sigma-Aldrich Inc., Saint Louis, MO, USA) that has been applied to induce kidney stones in several animal models. HLP can be metabolized to oxalate and induce calcium oxalate nephrolithiasis [34,35,36]. We coexposed Sprague–Dawley (SD) rats to either 63 mg/kg/day melamine or 126 mg/kg/day melamine (twice the NOAEL) with or without 2% HLP. Both chemicals were administered in drinking water.

The SD rats were obtained from BioLASCO Technology (Charles River Taiwan Ltd., Taipei, Taiwan) and were caged in an air-conditioned animal facility at 22 ± 1 °C and 50–70% humidity with a 12-h light/dark cycle. The rats were maintained with free access to water and a normal chow diet consisting of 11% fat (as a percentage of total kcal), 65% carbohydrate, and 24% protein (Maintenance Diet 1320, Altromin Spezialfutter GmbH & Co., KG, Lage, Germany). The concentrations of melamine in water were adjusted according the average amount of water that the rats consumed every day.

We first conducted a pilot study in which twelve male SD rats were randomly distributed into four groups: a control group (*n* = 3), a 2% HLP group (*n* = 3), a group receiving 63 mg/kg/day melamine + 2% HLP (*n* = 3), and a group receiving 126 mg/kg/day melamine + 2% HLP (*n* = 3). After obtained some preliminary result that demonstrated an increased incidence of calcium oxalate in urine of the rats coexposed to melamine with HLP (Appendix A), we conducted the second and third batches of experiments. In these two batches of experiments, a total of 49 male SD rats were randomly distributed into five groups: Group I (control, *n* = 10), Group II (126 mg/kg/day melamine, *n* = 9), Group III (2% HLP, *n* = 10), Group IV (63 mg/kg/day melamine + 2% HLP, *n* = 10), and Group V (126 mg/kg/day melamine + 2% HLP, *n* = 10). For Groups II, IV, and V, the rats were pre-exposed to melamine in drinking water for 2 weeks and then exposed to melamine only or coexposed to melamine and 2% HLP in drinking water for another 4 weeks. A scheme of the animal study is shown in Appendix A.

At the end of the experiment, the rats were individually transferred to metabolic cages for collecting 24 h urine. The urine was used for analyzing clinical biochemical data, oxidative stress markers and kidney injury markers. The rats were sacrificed after 6 weeks. After fasting overnight, the rats were anesthetized by intraperitoneal injection with Zoletil 50 (160 mg/kg) (Virbac, Carros, France). Blood samples were collected from the hearts at the time of sacrifice and centrifuged at 3000 rpm for 15 min. Then, the supernatant plasma was stored in a −20 °C freezer. Clinical biochemical data of the plasma and urine were measured by a Roche Cobas Integra 400 Chemistry Analyzer (Roche Diagnostics, Taipei, Taiwan).

### 2.7. Measurement of Oxidative Stress and Kidney Injury Biomarkers

Urinary 8-oxo-2′-deoxyguanosine (8-OHdG) and malondialdehyde (MDA) levels were measured with high-sensitivity 8-OHdG ELISA kits (cat. KOG-HS10E; JalCA, Fukuroi, Shizuoka, Japan) and thiobarbituric acid reactive substances (TBARS) assay kits (cat. 10009050; Cayman Chemical Inc., Ann Arbor, MI, USA), respectively. Urinary βeta-2-microglobulin (β2M; cat. SEA260Ra) and N-acetyl-β-D-glucosaminidase (NAG; cat. SEA069Mu) were measured with ELISA kits (USCN Life Science, Inc., Wuhan, China). Urinary oxalate was measured with Oxalate colorimetric assay kit (cat. K663; BioVision Ltd., Milpitas, CA, USA). The protocols were performed according to the kit suppliers’ instructions.

### 2.8. Detection of 8-OHdG and MDA in Kidney Tissues

To measure the levels of 8-OHdG and MDA in renal tissues, kidneys were homogenized in 1× PBS. Then, genomic DNA was extracted from a portion of the tissue homogenate with a PureLink™ Genomic DNA Mini Kit (cat. K182002; Thermo Fisher Scientific, Inc.) for 8-OHdG measurement using a high-sensitivity 8-OHdG ELISA kit. Another portion of tissue homogenate was lysed in M-PER™ Mammalian Protein Extraction Reagent (cat. 78501; Thermo Fisher Scientific, Inc.) for MDA measurement using a TBARS assay kit (Cayman Chemical Inc., Ann Arbor, MI, USA). The protocols followed the instructions provided by the kit suppliers.

### 2.9. Analysis of Kidney Morphology, Fibrosis, and Apoptosis

Kidneys were removed immediately after the rats were sacrificed. Formalin-fixed, paraffin-embedded kidney sections of 5 mm thickness were deparaffinized in xylene and rehydrated. Periodic acid–Schiff (PAS) staining with a PAS Stain kit (cat. 395B; Sigma-Aldrich, Inc., Saint Louis, MO, USA) was used to examine the histological morphology of the kidneys, whereas picrosirius red staining with a Picrosirius Red Stain kit (cat. Ab150681; Abcam Plc., Cambridge, MA, USA) was used to examine the severity of renal fibrosis. The protocols followed the instructions provided by the kit suppliers.

Apoptotic renal tubular cells were detected by terminal deoxynucleotidyl transferase dUTP-mediated nick-end labeling (TUNEL) assay using an In Situ Cell Death Detection kit, POD (cat. 11684817910; Roche Applied Science, Penzberg, Germany) according to the instructions provided by the kit supplier.

### 2.10. Immunohistochemistry for Oxidative Stress and Inflammatory Markers

A series of immunohistochemical (IHC) examinations with appropriate antibodies were used to examine biomarkers or histological changes in kidneys. An ED-1 antibody was used for macrophages, and an MDA antibody was used for lipid peroxidation. Nrf2 and DNA repair enzyme 8-oxoguanine DNA glycosylase 1 (OGG1) were detected with their specific antibodies. Immunohistochemical staining was performed according to the standard avidin-biotin-peroxidase complex method with an ABC Peroxidase Standard Staining Kit (cat. 32020; Thermo Fisher Scientific, Inc.). Oxidative damaged DNA was detected by an 8-OHdG antibody and immunohistochemical staining was performed with a VECTASTAIN ABC-AP kit (cat. AK-5002; Vector Laboratories, San Francisco, CA, USA) and a Vector Red Substrate kit (cat. SK-5100; Vector Laboratories). The information for the antibodies is listed in Appendix A.

### 2.11. Quantification of Tissue Images

Tubular luminal areas, fibrosis areas, macrophages, apoptotic cells, 8-OHdG staining, MDA staining, and Nrf2-positive nuclei were quantified in renal sections (six animals/group from the third batch experiment; 5–10 random fields per section). The collected images were analyzed under a light microscope and quantified using ImageJ software.

### 2.12. Protein Extraction and Western Blotting

To analyze the expression of the target proteins, HK-2 cells and the kidneys of rats from the second and third batch experiments were homogenized and lysed in M-PER Mammalian Protein Extraction Reagent supplemented with cOmplete™ Protease Inhibitor Cocktail (cat. 11697498001; Roche Applied Science, Penzberg, Germany). The cell nuclear proteins were extracted using a Nuclear Protein Extraction kit (cat. NPI-1; Fivephoton Biochemicals, San Diego, CA, USA). The protein concentrations were measured using a Pierce™ BCA Protein Assay kit (cat. 23227; Thermo Fisher Scientific, Inc.). For analysis of target proteins, equal amounts of cell or tissue lysates were loaded and separated on 7.5%, 10%, or 15% sodium dodecyl sulfate (SDS)-polyacrylamide gels. After transfer to polyvinylidene difluoride (PVDF) membranes, the proteins of interest were detected using corresponding antibodies. The purchase information for the antibodies is listed in Appendix A. The protein expression on the blot images was quantified by ImageJ software.

### 2.13. Statistics

The results for all groups are presented as the mean ± standard error (SE) and were analyzed with GraphPad Prism 9 software (GraphPad Software Inc., San Diego, CA, USA). One-way analysis of variance (ANOVA) followed by Tukey’s multiple comparison test was used to analyze the differences across groups. All *p*-values were two-sided with significance accepted at < 0.05.

## 3. Results

### 3.1. Diminishing Nrf2 Enhanced Oxidative Stress Induced by Melamine and Oxalate Coexposure in Human Proximal Tubular Cells

We first observed whether coexposure of melamine with SO could induce oxidative stress and influence the expression of Nrf2 and antioxidants in HK-2 cells. As shown in Figure 1A,B, melamine and SO coexposure induced more ROS production than melamine or SO single exposure. When Nrf2 was knocked down with a specific siRNA, the overall ROS production was further increased in the four groups, especially in the coexposure group.

We also investigated whether mitochondria are involved in the ROS generation induced by melamine and SO. As shown in Figure 1A,C, the fluorescence intensity of MitoTracker Red CMXRos was significantly increased in HK-2 cells exposed to melamine and/or SO, especially when Nrf2 was simultaneously knocked down. The increased fluorescence intensity indicated overproduction of mitochondrial superoxide anion since MitoTracker Red CMXRos is oxidized by superoxide, which results in the emission of red fluorescence [37].

### 3.2. Diminishing Nrf2 Blocked Antioxidative Enzymes Induced by Melamine and Oxalate Coexposure in Human Proximal Tubular Cells

In the single-exposure and coexposure groups, the protein expression of SOD1, catalase and HO1 was increased, possibly because of the translocation of Nrf2 into cell nuclei in response to oxidative stress (Figure 2A–G). To prove whether these antioxidant enzymes were regulated by Nrf2, we blocked Nrf2 with its siRNA. As shown in Figure 2A–C, siNrf2 treatment reduced 50% of Nrf2 total protein level and 30% of Nrf2 protein level in cell nuclei of the HK-2 cells. Nrf2 knockdown decreased the degrees of SOD1, catalase, and HO1 upregulation (Figure 2A–G). In contrast, the expression of the mitochondrial antioxidant SOD2 did not change in response to the increase in oxidative stress (Figure 2A,E). These results demonstrated that oxidative stress induced by melamine and/or oxalate exposure resulted in a complementary increase in antioxidative enzymes via Nrf2. Blockage of Nrf2 decreased antioxidative defense, which resulted in accumulation of substantial ROS induced by the exposure of melamine and/or oxalate.

### 3.3. Melamine and SO Coexposure Upregulated Markers of Inflammation, Apoptosis, and Fibrosis in Human Proximal Tubular Cells

Coexposure to melamine and SO induced significant apoptosis in HK-2 cells (Figure 3A,B). Nrf2 knockdown further increased HK-2 cell apoptosis in the single-exposure and coexposure groups, indicating that programmed cell death may, at least in part, be caused by oxidative stress (Figure 3A,B). As shown in Figure 3C–E, translocation of NF-κB to HK-2 cell nuclei was significantly increased in coexposure group, and the translocation was further increased by Nrf2 knockdown in the control, single-exposure, and coexposure groups. The expression of collagen IV, an extracellular matrix protein, was significantly increased by Nrf2 knockdown in HK-2 cells coexposed to melamine and SO (Figure 3C,F). The protein levels of cleaved caspase-3, which is responsible for executing apoptosis, were found to be significantly elevated in the coexposed HK-2 cells (Figure 3C,G). Taken together, our results suggest that coexposure to melamine and oxalate can enhance inflammation, apoptosis and fibrosis in kidney proximal tubular cells.

### 3.4. Melamine and Oxalate Coexposure Enhanced Lethal and Sublethal Renal Tubular Damage

Increased 24-h urinary oxalate excretion has been reported to be a risk factor for kidney stone formation and CKD [3,31]. In the rat model, we observed an elevated frequency and incidence of calcium oxalate crystals in urinary specimens from the coexposed groups (Appendix A). In addition, urinary oxalate excretion was elevated in the HLP single-exposure group and the coexposure groups (Appendix A). These results imply that melamine may accelerate the formation of calcium oxalate crystals or stones in the kidneys.

To clarify whether the coexposure can cause kidney damage or kidney stones, we observed the kidney morphology. As shown in Figure 4A,B, tubular luminal areas were significantly increased in the kidneys of the coexposed rats, suggesting the induction of renal tubular dilation, an early sign of kidney damage. In some of the dilated renal tubules, we observed loss of brush borders and shedding of cell nuclei (Figure 4A). The rats coexposed to the high dose of melamine (126 mg/kg/day) and HLP exhibited the most pronounced renal tubular damage, such as tubule atrophy and interstitial infiltration of chronic inflammatory cells (Figure 4A). However, stone formation in the renal tubular areas was not observed.

Loss of renal tubular cells could be the cause of tubular atrophy. We found that apoptotic cells were significantly increased in the kidneys of the coexposed rats than in those of the rats exposed to HLP or melamine alone (Figure 4C,D). It is noteworthy that apoptosis was present mainly in the renal tubules (Figure 4C). Severe apoptosis leading to tubular atrophy was observed in some regions of the kidneys in the high-dose melamine coexposure group (Figure 4C).

To observe the effect of melamine and oxalate coexposure on kidney function, we measured some clinical biochemical parameters and kidney injury markers in plasma and urine. The results showed that commonly measured biochemical parameters and biomarkers, such as blood urea nitrogen (BUN), creatinine, β2M, and NAG, were similar across all five groups (Appendix A). Even though these biochemical parameters did not show a significant decline in kidney function, the morphological changes of kidneys in the coexposed rats indicated that early signs of kidney injury have occurred.

### 3.5. Melamine and Oxalate Coexposure Increased Renal Tubulointerstitial Macrophage Infiltration and Fibrosis

Inflammation is a critical mediator of kidney damage. As shown in Figure 5A,B, macrophage infiltration in the renal tubular interstitium was significantly increased in the rats coexposed to melamine and HLP compared to those exposed to melamine or HLP alone. The results indicate that inflammation may be involved in the early kidney injury caused by melamine and oxalate coexposure.

Renal fibrosis is the main pathological basis of CKD progression to ESRD. We used picrosirius red staining to observe fibrosis and found that the accumulation of extracellular matrix proteins in the renal tubular interstitium was significantly elevated (by approximately 2–4%) in the coexposed rats (Figure 5C,D). The results indicate that the coexposure can also cause early kidney fibrosis in rats.

### 3.6. Melamine and Oxalate Coexposure Upregulated Expression of Proteins Involving Kidney Injury

We also observed protein levels of several molecules that have been reported as kidney injury markers or regulators participating in the processes of inflammation, fibrosis and apoptosis (Figure 6A,B). We found that the ratio of phosphorylated NF-κB to total NF-κB levels was significantly increased in the kidneys of melamine single exposed and melamine-oxalate coexposed groups. In addition, the protein expression levels of both phosphorylated NF-κB and total NF-κB were remarkably increased in the kidneys of the coexposed rats.

The protein level of kidney injury molecule-1 (KIM-1), an early kidney tubular injury marker [38], was increased in the four exposed groups, especially the high-dose melamine and oxalate coexposed group. TGF-β1 protein levels in the kidneys were significantly upregulated in the four exposed groups and the level in the high-dose melamine and oxalate coexposed group was the highest. Additionally, protein expression of collagen IV was upregulated in the HLP single exposed and the melamine-oxalate coexposed groups.

Apoptosis can be induced by Bax-dependent caspase-3 activation [39]. Our results demonstrated that Bax and caspase-3 protein levels were increased in the kidneys of the four exposed groups. Similarly, these two proteins were expressed the highest levels in the high-dose melamine and oxalate coexposed group. The ratio of cleaved-caspase-3 to caspase-3, indicating an activation of caspase-3, was significantly increased only in the high-dose melamine and oxalate coexposed group.

### 3.7. Melamine and Oxalate Coexposure Induced Oxidative Stress in the Kidneys

To clarify whether oxidative stress was a key player involving in the early kidney injury induced by melamine and oxalate coexposure, we observed MDA levels in the kidneys and urine. As shown by IHC staining, the coexposure groups presented stronger MDA intensity than the control and single-exposure groups (Figure 7A,B), especially in the damaged renal tubules (Figure 7A). MDA levels in kidney tissues were significantly higher in the rats coexposed to melamine and HLP than in those exposed to HLP or melamine alone (Figure 7C,D). In addition, urinary MDA concentrations were significantly increased in the coexposure groups (Figure 7E).

### 3.8. Melamine and Oxalate Coexposure Caused DNA Oxidative Damage in the Kidneys

We also observed the oxidative DNA damage marker 8-OHdG. The intensity of 8-OHdG staining was substantially stronger in the kidney sections of the coexposure groups, especially in the damaged renal tubules, than in those of the control and single-exposure groups (Figure 8A–C). Specifically, we observed that more 8-OHdG accumulated in the cell nuclei of the damaged renal tubules in the high-dose melamine coexposure group than in the other groups (Figure 8A). Consistently, DNA extracted from kidney tissues of the rats coexposed to both melamine and HLP presented significantly higher quantities of 8-OHdG than those of the rats exposed to HLP or melamine alone (Figure 8D). Urinary 8-OHdG concentrations in the coexposure groups were also higher than those in the control and single-exposure groups (Figure 8E).

### 3.9. Melamine and Oxalate Coexposure Reduced Nrf2 and OGG1 Translocation to Nuclei of Damaged Renal Tubular Cells

The kidney levels of Nrf2 protein were significantly upregulated in the low-dose melamine coexposure group and in the HLP and melamine single-exposure groups compared with the control group (Figure 9A,B). In the high-dose melamine coexposure group, Nrf2 protein level in the kidney was upregulated but did not reach a statistic significance when compared with the control group (Figure 9A,B). Observed by IHC, Nrf2 was present at nuclei of renal tubular cells in the normal control rats (Figure 9D). Notably, our results demonstrated that Nrf2 was not sufficiently translocated into cell nuclei of the damaged renal tubules in both low-dose and high-dose melamine coexposure rats (Figure 9D,E). These results suggest that increased expression of Nrf2 may not trigger satisfactory antioxidative defense if it could not be successfully translocated into cell nuclei.

OGG1 is a DNA repair enzyme that is present in cell nuclei and mitochondria. As shown in Figure 9A,C, OGG1 proteins were significantly upregulated in the kidneys of low-dose melamine coexposure group and HLP and melamine single-exposure groups compared with the control group. In contrast, OGG1 protein level was not changed in the kidneys of high-dose melamine coexposure group (Figure 9A,C). According to the IHC results (Figure 9F), OGG1 was present at cell nuclei of the renal tubules in normal control rats. Similar to Nrf2, OGG1 could not be sufficiently transported into cell nuclei of the damaged renal tubules in the melamine and oxalate coexposed rats (Figure 9F), which may have impeded the repair of oxidative DNA damage. Thus, accumulation of 8-OHdG in cell nuclei of the damaged renal tubules perhaps caused by the OGG1 enzyme that was not sufficiently produced or successfully delivered into cell nuclei to remove 8-OHdG.

### 3.10. Melamine and Oxalate Coexposure Influenced Expression of Antioxidative Enzymes

The protein levels of several antioxidants in the kidneys were also measured (Figure 10A,B). The protein levels of NQO1, SOD1, and HO1 showed no difference among the five groups. Kelch-like ECH-associated protein 1 (Keap1), SOD2, and catalase protein levels were significantly upregulated in the low-dose melamine coexposure group and in the HLP and melamine single-exposure groups compared with the control group. In contrast, the kidney levels of these proteins were decreased or were not sufficiently upregulated in the high-dose melamine coexposure group. These results indicate that coexposure to high-dose melamine and HLP may lead to insufficient upregulation of stress-defense regulators and antioxidants, resulting in further renal tubular injury.

## 4. Discussion

Humans are commonly exposed to mixtures of numerous trace chemicals in their daily environments. It still needs more efforts to investigate the context of complex chemical interactions in the pathology of CKD and ESRD. Oxalate abundantly exists in many food products and can be easily ingested into human bodies. Melamine is considered a relatively safe chemical and is widely present in many materials used in daily life. In this study, we explored whether the NOAEL of melamine remains harmless when oxalate is also present. Our results suggest that coexposure to NOAEL of melamine with oxalate can cause greater oxidative stress in the kidneys, particularly in the renal tubular region, than melamine or oxalate single exposure. The morphological observations indicate that the degree of renal tubular injury is associated with oxidative stress intensity and repair ability. Thus, carefully avoiding the nephrotoxicity induced by the interaction of these two common environmental chemicals should be advised.

We have reported that increased urinary melamine levels are positively associated with renal tubular injury and kidney function decline [20,30,40], but the underlying mechanism by which environmental exposure of melamine can cause kidney injury remains unclear. The formation of calcium oxalate kidney stones, the most common type of kidney stones in the population, is also a cause of CKD progression [3,5]. Hyperoxaluria has been considered one of the major risk factors for developing calcium oxalate kidney stones [4,34,41]. In this study, we established a coexposure scenario where the two environmental potential nephrotoxicants were examined for their synergistic effects on kidney injury. Particularly, in the in vitro study, we used a melamine concentration within the range detected in the urine of melamine manufacturers [30]. The concentration of oxalate used to treat the cells in this study is also lower than the urinary oxalate concentration that has been reported to increase the risk of kidney stone formation and CKD [3,31]. Our results demonstrate that at these low concentrations of melamine and oxalate, only the coexposure could stimulate human kidney proximal tubular cells to significantly increase production of ROS, translocation of the inflammatory transcriptional factor NF-κB to cell nuclei, and apoptosis. We also observed similar results in the in vivo study. Exposure to high-dose melamine (twice the NOAEL) alone did not cause substantial kidney damage. Although HLP single exposure caused significant renal tubular dilation, the degree of inflammation and kidney injury remained relatively low. In contrast, compared to melamine or HLP single exposure, combined exposure to the NOAEL of melamine and HLP resulted in markedly more severe kidney damage, including greater inflammation, oxidative DNA damage, tubular cell apoptosis, tubular atrophy, and interstitial fibrosis. Together, both in vitro and in vivo studies demonstrate that melamine and oxalate have a synergistic effect of nephrotoxicity.

Oxidative damage is clinically important and has been reported to play a significant pathological role in many diseases, including malignancy, atherosclerosis, hypertension, ischemic heart disease, and CKDs. Kidneys are highly susceptible to oxidative stress, which is regarded as a critical pathogenic step in the initiation, development, and/or progression of most types of CKDs [25,29,42,43,44,45]. Oxidative stress occurs when the ROS generation exceeds the endogenous antioxidant capacity. Our previous in vitro study demonstrated that exposure to melamine alone can induce inflammation and oxidative stress, which can further result in inflammatory and fibrogenic protein production and renal proximal tubular cell apoptosis [21]. Recently, we further revealed the interrelationships of environmental melamine, oxidative stress, and early kidney injury in human studies [22]. However, the detailed mechanisms by which mediate the renal tubular injury caused by environmental concentrations of melamine through oxidative stress remain elusive. Some studies have found that in response to oxidative stress, the transcription factor Nrf2 can be activated and induce structural changes in Keap1, which allows nuclear translocation of Nrf2 and upregulates the expression of antioxidant enzymes, such as NQO1, catalase, HO1, SOD1, and SOD2 [26,27,28,29,45]. Nrf2 was reported to be downregulated or inactivated in several CKD animal models, including the indoxyl sulfate (a uremic toxin) treated rats [46], adenine-diet induced CKD rats [47], and db/db diabetic mice [48]. Thus, Nrf2 has been suggested as a therapeutic target for CKD [45]. As we shown in this study, after HK-2 cells were stimulated with melamine and/or SO, translocation of Nrf2 into cell nuclei was increased; this increase at least partly influenced upregulation of catalase, HO1, and SOD1, which prevented the cells from producing large quantities of ROS and protected the cells from injury. When Nrf2 was knocked down in HK-2 cells, ROS levels were dramatically increased by exposure to melamine and/or SO, which might have resulted in massive translocation of NF-κB into nuclei, production of collagen IV, cleavage of caspase 3, and apoptosis. The results indicate that Nrf2 may play a critical role in defense against injury induced by melamine and oxalate.

The current animal study also demonstrated that the protein levels of Nrf2, Keap1, SOD2, catalase, and OGG1 were significantly increased in the melamine or HLP single-exposure groups as well as in the coexposure groups treated with the NOAEL of melamine. In contrast, the protein levels of Nrf2, Keap1, SOD2, catalase and OGG1 in the coexposure group treated with double the NOAEL dose of melamine were not sufficiently increased or were even decreased. These results indicate that the defense against oxidative stress remained satisfactory in the single-exposure and the NOAEL of melamine coexposure conditions. However, when the NOAEL dose of melamine was doubled or the exposure period was extended, the defense system was exhausted and unable to respond to the harm caused by these toxicants, which eventually led to severe renal tubular damage. The mechanism may have involved a lack of catalase to convert hydrogen peroxide into oxygen and water after SODs converted superoxide radicals into hydrogen peroxide. Furthermore, oxidative DNA damage is a major cause of cell death. The most common product of DNA oxidation is the base lesion 8-OHdG, which is repaired by OGG1 to initiate the base excision repair pathway [49]. Our results demonstrated that 8-OHdG levels in renal tubules were significantly increased in the coexposure groups. In addition, decreases in OGG1 protein levels resulted in accumulation of 8-OHdG in cell nuclei and caused severe renal tubular damage in the coexposure group treated with double the NOAEL of melamine.

It is noteworthy that insufficient antioxidant defense may also cause by inadequate transportation of Nrf2 and OGG1 into cell nuclei to mediate the transcription of antioxidant genes and to excise the base lesion 8-OHdG, respectively. We observed that Nrf2 nuclear translocation was increased in HK-2 cells, which is in contrast to the animal study. This phenomenon may be due to the limitation between these two models in which the final concentrations of melamine and oxalate exposed to kidney cells and the time periods of exposure were different. These differences may explain why Nrf2 nuclear translocation was increased in HK-2 cells but decreased in renal tubules of the rats. Nezu and Suzuki reported that Nrf2 level was shortly increased in response to the oxidative stress induced by renal ischemia-reperfusion injury but returned to basal level within 24 h, indicating the Nrf2 activation was insufficient to eliminate long-term cytotoxic stress [50,51]. In our HK-2 cell model, the cells were only treated with melamine and oxalate for 6 h. This time period probably remained at the self-defense stage in response to oxidative stress, so increased Nrf2 nuclear translocation was observed in the HK-2 cells. The mechanism by which coexposure to melamine and oxalate impaired oxidative stress defense still needs further investigation.

One limitation of this study is that the clinical biochemical parameters related to kidney function did not show significant changes in the animal experiments. However, in this 6-week short term melamine/oxalate coexposure model, we still demonstrated significant kidney tubular cell injury that was not observed in a single-exposure model in which the rats were exposed to melamine at a dose of 63 mg/kg/day for 13 weeks [32,33]. In addition, extra groups of coexposed rats treated with Nrf2 activators or antioxidants could be added to further clarify the role of Nrf2 and oxidative stress. Particularly, it needs more investigation to explain how melamine and oxalate coexposure can prohibit Nrf2 and OGG1 to be transported into the cell nuclei of renal tubular cells.

## 5. Conclusions

Our study provides evidence that exposure to the NOAEL of melamine with other potential nephrotoxicants, such as oxalate from oxalate-rich foods, can lead to early kidney injury via oxidative stress. Diminishment of Nrf2 biological function results in lessened capacity of antioxidative defense, by which augments ROS generation and aggravates renal tubular cell damage caused by the melamine and oxalate coexposure.

## Figures and Tables

**Figure 1 antioxidants-10-01464-f001:**
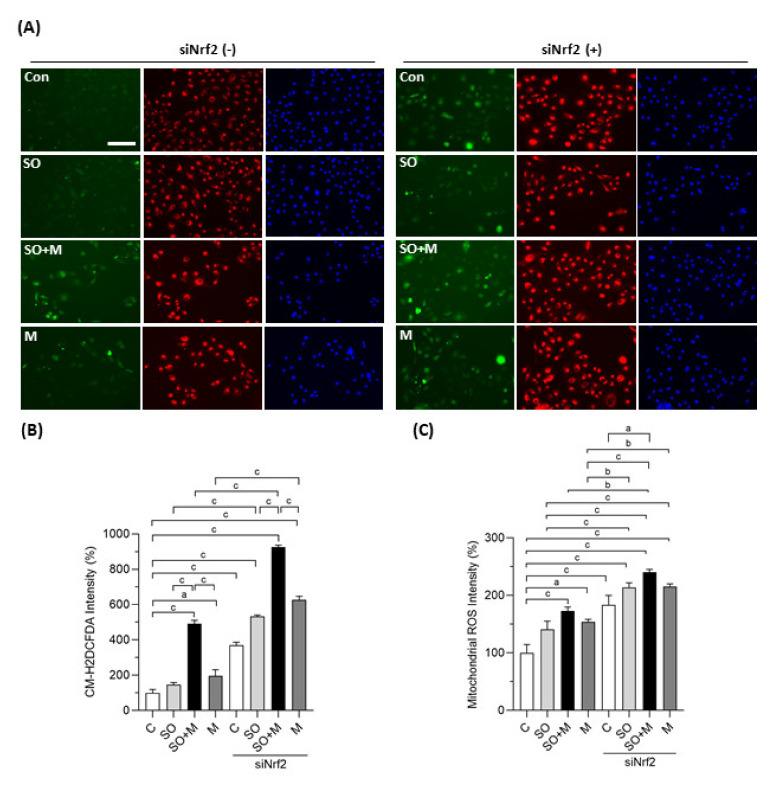
Cellular and mitochondrial ROS generation in HK-2 cells. (**A**) Production of cellular reactive oxygen species (ROS) in human proximal tubular cells (HK-2 cells) was detected by a CM-H_2_DCFDA green fluorescence probe (green color). Mitochondrial ROS was detected by a MitoTracker Red CMXRos fluorescent probe (red color). Cell nuclei were stained with Hoechst 33342 (blue color). Scale bar = 150 μM. (**B**) Quantified results of the cellular ROS. (**C**) Quantified results of the mitochondrial ROS. The cells were stimulated with 1 μg/mL melamine (M) and/or 0.01 mM sodium oxalate (SO) for 6 h. Con and C: control cells without treatment; siNrf2: the cells treated with Nrf2 small interference RNA for 24 h. Data are mean ± SE (*n* = 3). a: *p* < 0.05; b: *p* < 0.01; c: *p* < 0.001.

**Figure 2 antioxidants-10-01464-f002:**
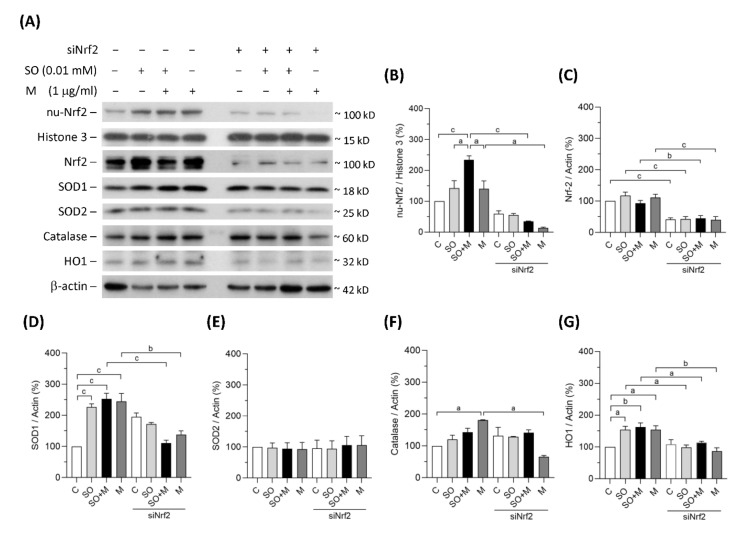
Protein expression of Nrf2 and antioxidants in HK-2 cells. (**A**) The representative immunoblot images for nuclear proteins of Nrf2 (nu-Nrf2) and histone 3, and total proteins of Nrf2, SOD1, SOD2, catalase, HO1, and β-actin. These protein expressions were detected by western blotting. (**B**–**G**) The quantified results of the blots. The cells were stimulated with 1 μg/mL melamine (M) and/or 0.01 mM sodium oxalate (SO) for 6 h. C: control cells without treatment; siNrf2: the cells treated with Nrf2 small interference RNA for 24 h. Data are mean ± SE (*n* = 3). a: *p* < 0.05; b: *p* < 0.01; c: *p* < 0.001.

**Figure 3 antioxidants-10-01464-f003:**
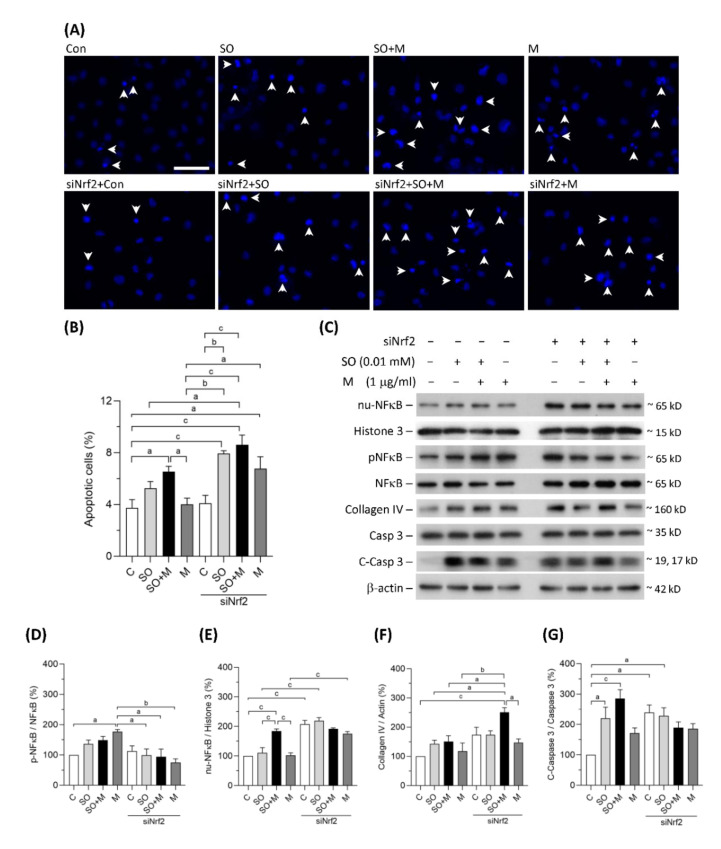
Apoptosis and the expression of proteins related to inflammation, fibrosis, and apoptosis in HK-2 cells. (**A**) Apoptotic HK-2 cells were detected by a Hoechst 33342 fluorescent probe. The apoptotic cells were indicated by white arrowheads. Scale bar = 75 μM. (**B**) Quantified results of the apoptotic cells. (**C**) The representative immunoblot images for nuclear proteins of NF-κB (nu-NFκB) and histone 3, phosphorylated protein of NF-κB (pNFκB), and total proteins of NF-κB, collagen IV, cleaved-caspase 3 (c-Caspase 3), caspase 3, and β-actin. The protein expressions were detected by western blotting. (**D**–**G**) The quantified results of the blots. The cells were stimulated with 1 μg/mL melamine (M) and/or 0.01 mM sodium oxalate (SO) for 24 h. Con and C: control cells without treatment; siNrf2: the cells treated with Nrf2 small interference RNA for 24 h. Data are mean ± SE (*n* = 3). a: *p* < 0.05; b: *p* < 0.01; c: *p* < 0.001.

**Figure 4 antioxidants-10-01464-f004:**
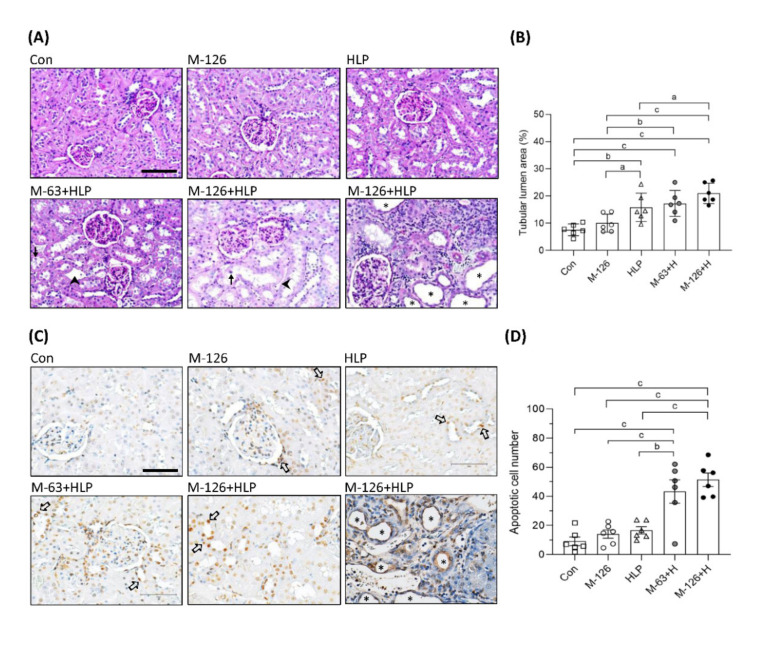
Morphological change and cell apoptosis in kidneys of the SD rats. (**A**) Representative images of periodic acid–Schiff (PAS) staining in kidneys of the rats. → indicates shedding of cell nuclei. ➤ indicates interstitial infiltration of inflammatory cells. Scale bar = 100 μM. (**B**) Quantified results for the renal tubular lumen area from the images of PAS staining. (**C**) Apoptotic cells in kidneys of the rats detected by TUNEL assay. The apoptotic cells were stained with DAB substrate (Brown color). ⇨ indicates some representative apoptotic cells in renal tubules. Scale bar = 75 μM. (**D**) Quantified results for the apoptotic cells. Cell nuclei were stained with hematoxylin. Stars indicate atrophy of renal tubules. Con: control; HLP or H: 2% hydroxy-L-proline; M-126: melamine 126 mg/kg/day; M-63: melamine 63 mg/kg/day. Data are mean ± SE from six rats of each group in the third batch animal experiment. a: *p* < 0.05; b: *p* < 0.01; c: *p* < 0.001.

**Figure 5 antioxidants-10-01464-f005:**
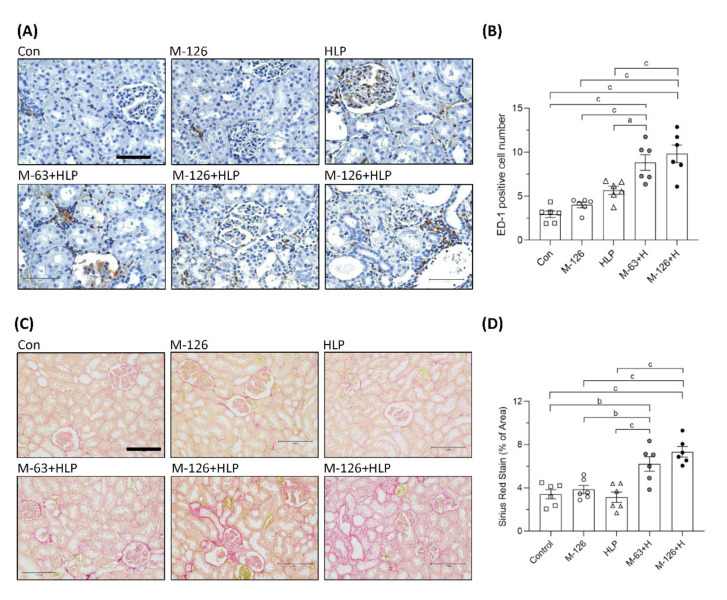
Inflammation and fibrosis in kidneys of the SD rats. (**A**) ED-1 IHC staining in kidneys of the rats. Brown color (DAB substrate) stains indicate macrophages (ED-1 positive stained cells) infiltration to renal tubulointerstitium, suggesting chronic inflammation occurred in kidneys of the rats coexposed to melamine and HLP. Scale bar = 75 μM. (**B**) Quantified results for the ED-1 positive stains. (**C**) Representative images of Picrosirius Red staining in kidneys of the rats. Red color stains in the kidneys indicate that positive staining of extracellular matrix proteins accumulate in tubulointerstitium, suggesting occurrence of renal fibrosis. Scale bar = 150 μM. (**D**) Quantified results for the fibrosis degree (red color) in the kidneys. Con: control; HLP or H: 2% hydroxy-L-proline; M-126: melamine 126 mg/kg/day; M-63: melamine 63 mg/kg/day. Data are mean ± SE from six rats of each group in the third batch animal experiment. a: *p* < 0.05; b: *p* < 0.01; c: *p* < 0.001.

**Figure 6 antioxidants-10-01464-f006:**
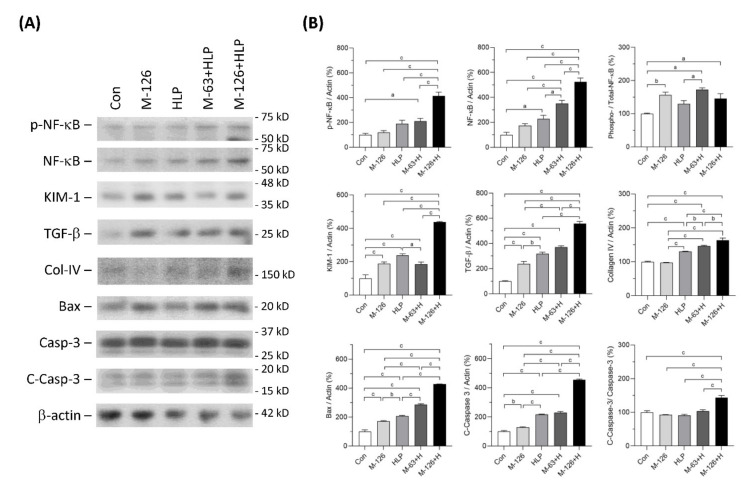
Expression of the proteins related to inflammation, fibrosis, and apoptosis in kidneys of the SD rats. (**A**) Representative immunoblots of phosphorylated NF-κB (pNFκB), total NF-κB, KIM-1, TGF-β, collagen IV (Col-IV), Bax, caspase-3 (Casp-3), cleaved-caspase-3 (C-Casp-3) and β-actin, detected by western blotting. (**B**) Quantified results for the immunoblots. Con: control; HLP or H: 2% hydroxy-L-proline; M-126: melamine 126 mg/kg/day; M-63: melamine 63 mg/kg/day. Data are mean ± SE from six rats of each group in the third batch animal experiment. a: *p* < 0.05; b: *p* < 0.01; c: *p* < 0.001.

**Figure 7 antioxidants-10-01464-f007:**
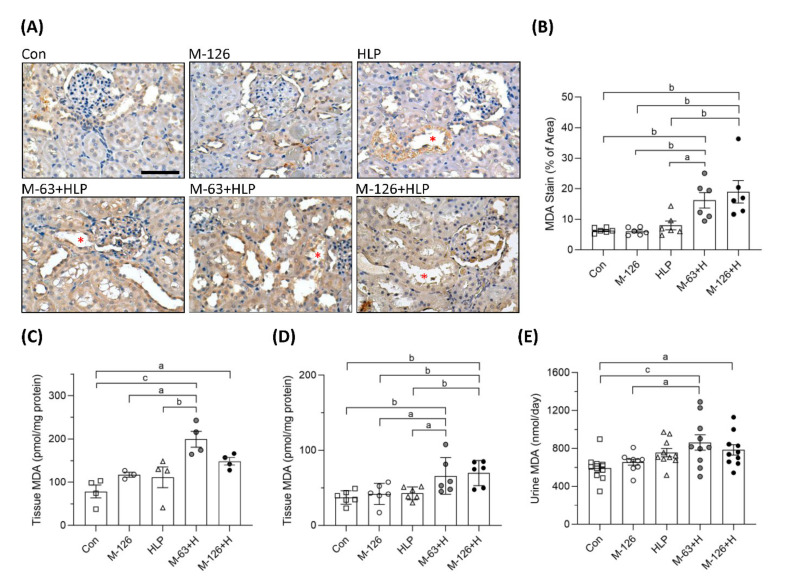
The levels of oxidative stress biomarker MDA in kidney tissues and urine of the SD rats. (**A**) Representative images of malondialdehyde (MDA) immunohistochemical (IHC) staining. Brown color indicates positive staining of MDA detected with 3′,3′-diaminobenzendine (DAB) substrate, suggesting that lipid peroxidation is increased in the kidneys. Red stars (*) indicate that stronger MDA stains present in the kidney tubules to be destroyed. Cell nuclei were counterstained with hematoxylin. Scale bar = 75 μM. (**B**) Quantified results of the MDA stains in the IHC images (*n* = 6 for each group). (**C**) MDA levels in kidney tissues from the second batch animal experiment (*n* = 3–4 for each group) measured by ELISA. (**D**) MDA levels in kidney tissues from the third batch animal experiment (*n* = 6 for each group) measured by ELISA. (**E**) Urinary MDA levels measured by ELISA (*n* = 9–10 for each group). Con: control; HLP: 2% hydroxy-L-proline; Mel-126: melamine 126 mg/kg/day; Mel-63: melamine 63 mg/kg/day. Data are mean ± SE. a: *p* < 0.05; b: *p* < 0.01; c: *p* < 0.001.

**Figure 8 antioxidants-10-01464-f008:**
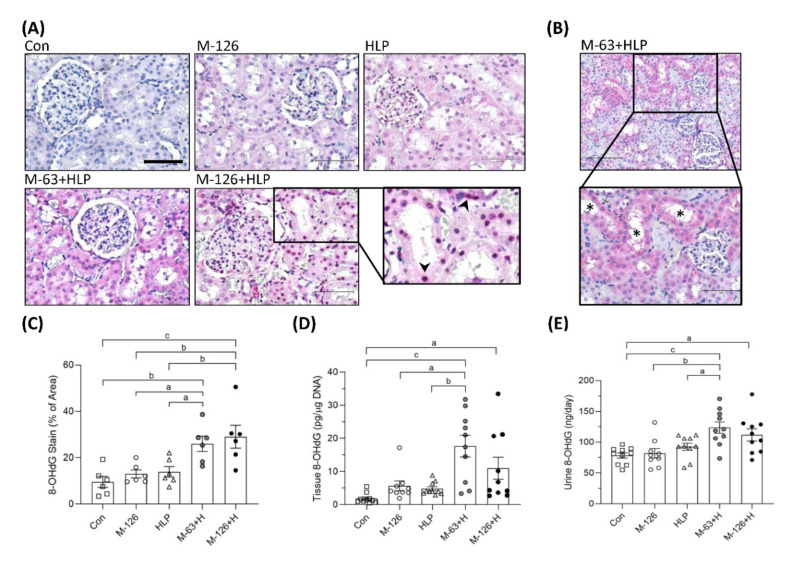
DNA oxidative damage in kidneys and urine of the SD rats. (**A**) Representative images of 8-OHdG IHC staining. Pink and red colors indicate positive stains of 8-OHdG detected with Vector^®^ Red substrate, suggesting that oxidative DNA damage occurred in the kidneys. Cell nuclei were counterstained with hematoxylin. Scale bar = 75 μM. Arrowheads (➤) indicate that 8-OHdG is accumulated in cell nuclei of the damaged renal tubules. (**B**) An representative image with lower magnification power (100×). Stars (*) indicate that damaged renal tubules present stronger positive 8-OHdG stains. (**C**) Quantified IHC results of the 8-OHdG positive stains (*n* = 6 for each group). (**D**) The levels of 8-OHdG in the kidney tissues measured by ELISA (*n* = 9–10 for each group). (**E**) Urinary 8-OHdG levels measured by ELISA (*n* = 9–10 for each group). Con: control; HLP: 2% hydroxy-L-proline; Mel-126: melamine 126 mg/kg/day; Mel-63: melamine 63 mg/kg/day. Data are mean ± SE. a: *p* < 0.05; b: *p* < 0.01; c: *p* < 0.001.

**Figure 9 antioxidants-10-01464-f009:**
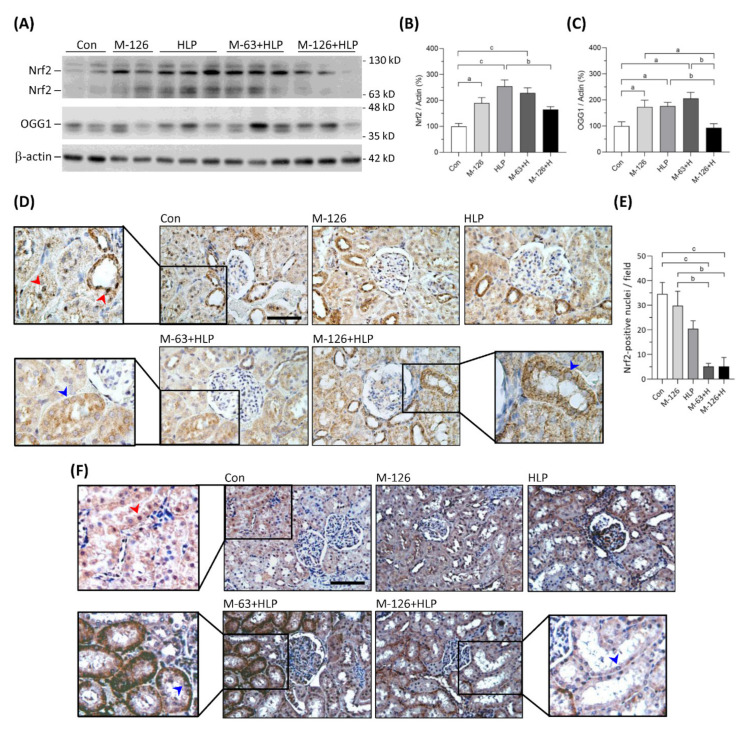
Protein expression of Nrf2 and OGG1 in kidneys of the SD rats. (**A**) Representative images of the immunoblots. The protein levels of Nrf2, OGG1 and β-actin were detected by western blotting. (**B**,**C**) Quantified results for the western blotting (*n* = 9–10 for each group). (**D**) Representative images of Nrf2 IHC staining. (**E**) Quantified results of Nrf2 presented in the cell nuclei of renal tubules (*n* = 6 for each group). (**F**) Representative images of OGG1 IHC staining. Brown color indicates positive stains of Nrf2 or OGG1 proteins detected with DAB substrate. Red arrowheads indicate that Nrf2 or OGG1 are present in cell nuclei of the renal tubules. Blue arrowheads indicate that Nrf2 or OGG1 are not able to translocate in cell nuclei of the damaged renal tubules. Con: control; HLP: 2% hydroxy-L-proline; Mel-126: melamine 126 mg/kg/day; Mel-63: melamine 63 mg/kg/day. Data are mean ± SE. a: *p* < 0.05; b: *p* < 0.01; c: *p* < 0.001.

**Figure 10 antioxidants-10-01464-f010:**
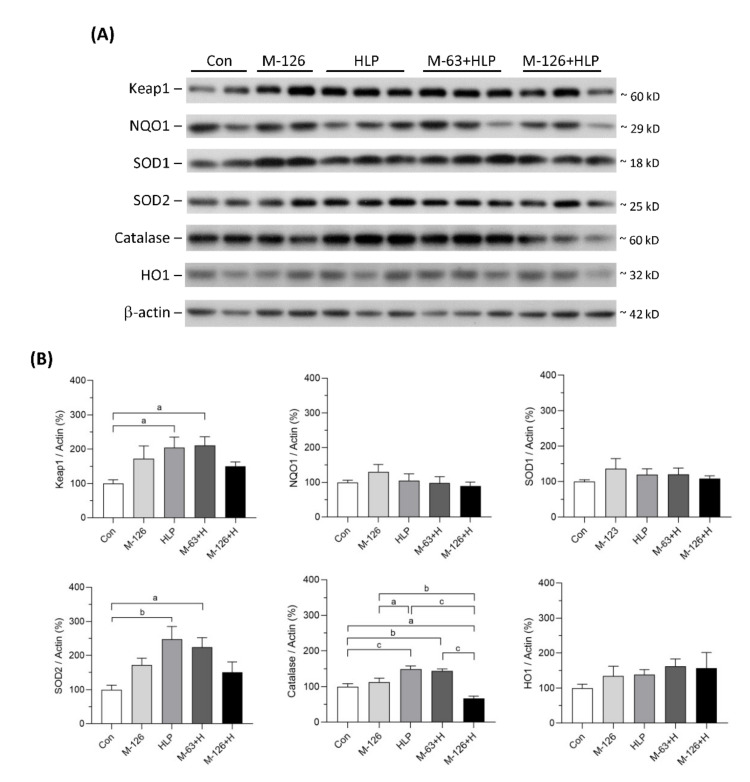
Protein expression of stress-defense regulators and antioxidants in kidneys of the SD rats. (**A**) Representative images of the immunoblots. The protein levels of Keap1, NQO1, SOD1, SOD2, catalase, HO1, and β-actin were detected by western blotting. (**B**) The bar charts are quantified results for the immunoblots of the target proteins. Con: control; HLP or H: 2% hydroxy-L-proline; M-126: melamine 126 mg/kg/day; M-63: melamine 63 mg/kg/day. Data are mean ± SE from 9–10 rats of each group from the second and third batch experiments. a: *p* < 0.05; b: *p* < 0.01; c: *p* < 0.001.

## Data Availability

The data is contained within this article or supplementary file.

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
