# Peer review of "Diminishment of Nrf2 Antioxidative Defense Aggravates Nephrotoxicity of Melamine and Oxalate Coexposure"

_antioxidants, 2021, doi:10.3390/antiox10091464_

Round 1

Reviewer 1 Report

Major comments.

The present study focused on oxidative stress as a mechanism for the enhancement of renal damage by melamine and oxalate and further suggested that the diminishment of Nrf2 was associated with exacerbation. In cell culture experiments, the authors demonstrated that Nrf2 have a crucial role in the toxicity of melamine and oxalate. Furthermore, co-administration experiments indicated nephrotoxicity aggravation in rats. In conclusion, the authors suggested that reduction or incomplete induction of Nrf2 exacerbates renal injury through disruption of anti-oxidative mechanisms, but why such phenomena were induced only by high-dose was unclear. It is unclear whether high-dose administration results in (1) suppression of Nrf2 activation or (2) depletion or inactivation following Nrf2 activation (overshoot). In animal experiments, it can be explained by evaluation over time, but since it takes many times, it may be better to explain it at least in cell experiments, etc., or it is better to explain it by referring to the literature appropriately.The animal studies in Figure 4-10 well illustrated the exacerbation of renal damage by melamine and oxalate. It seems to be necessary to examine by cell experiment or the discussion in order to explain these mechanisms.

Methods, cell culture.

In general, HK-2 cells are recommended to be cultured in keratinocyte-serum free medium with 5 ng/ml recombinant epidermal growth factor and 50 ug/ml bovine pituitary extract. Do the authors confirm the presence of a tubular epithelial-like phenotype in this condition compared to recommended condition? I know that the authors used 10% FBS containing DMEM in a previously published study (Ref. 21).

Figures 2 and 9.
In animal experiments, high-dose co-administration increased the expression of Nrf2 but decreased its nuclear localization, whereas, in cell experiments, the nuclear localization of Nrf2 was increased. If high doses suppress nuclear translocation of Nrf2 in animal studies, is it confirmed that HK-2 cells are similarly induced by the addition of excessive amounts of SO + M? Please discuss the differences between the two experiments, or referred and discuss molecular mechanisms by which Nrf2 nuclear translocation is reduced.

Figure 3.

Although apoptotic cells increased in siNrf2-treated SO, SO+M, or M, but not siNrf2 control. However, c-casp3 expression remained unchanged in SO, SO + M, and M, and c-casp3 expression increased considerably in the control of siNrf2. These results suggest that the decrease in Nrf2 may not enhance the toxicity of M and SO.

Author Response

We thank reviewer's constructive comments. The responses to comments are attached in the PDF file. 

Reviewer 2 Report

The report describes the results of cell culture and animal studies showing nephrotoxic effects of simultaneous exposure to oxalate and melamine. The study is a continuation of previous research of the authors regarding melamine toxicity. In my opinion, the report is informative and clearly written. There are several minor issues that should be corrected.

  1. Please clarify the difference between batch 2 and batch 3 animal experiment and the reason for repeating the experiment. Could you include data from batch 2 experiment e.g. in Figure 2? In addition, the numbers of animals in batch 2 experiment denoted in Figure S1 and Table S1 are not the same.
  2. The text contains several grammar or typing errors, e.g. in lines 117-120, line 350, line 463, line 480, several sentences in Discussion, the legend of the last supplementary figure, line 946. Line 513: please correct “starts” into “stars”; the same mistake is repeated in captions to Figure 7 and 8. Please delete the fragments of text regarding the requested manuscript format (lines 48-56; 831-834).
  3. Line 456: Table S2 should be cited instead of S3.
  4. Line 853: The word “human” should be avoided as the study describes cell culture and animal experiments.

Author Response

(The authors gave the same response as above.)

Round 2

Reviewer 1 Report

The manuscript has been improved well, and I think this manuscript will be acceptable.

Author Response

We thanks reviewer's comments.